# Integrative Multi-Omics Analyses Reveal Mechanisms of Resistance to Hsp90β-Selective Inhibition

**DOI:** 10.3390/cancers17213488

**Published:** 2025-10-30

**Authors:** Ian Mersich, Eahsanul Anik, Aktar Ali, Brian S. J. Blagg

**Affiliations:** 1Department of Chemistry and Biochemistry, University of Notre Dame, Notre Dame, IN 46556, USA; imersich@nd.edu (I.M.); manik@nd.edu (E.A.); aali4@nd.edu (A.A.); 2Warren Family Research Center for Drug Discovery and Development, University of Notre Dame, Notre Dame, IN 46556, USA

**Keywords:** Hsp90, HSP90AB1, multi-omics, network analysis, pharmacologic resistance, kynurenine, dependency

## Abstract

**Simple Summary:**

Isoform-selective inhibition of the molecular chaperone Hsp90β represents a promising anticancer strategy with improved safety relative to pan-Hsp90 inhibitors. Through integrative multi-omics analyses across hundreds of cancer cell lines and validation with Hsp90β-selective compounds, this study defines molecular mechanisms underlying resistance to Hsp90β inhibition. We show that resistant cells engage compensatory stress response and cytoskeletal remodeling programs, as well as metabolic adaptation involving kynurenine-mediated AHR activation. Importantly, these mechanisms reveal therapeutic opportunities, including synergy between Hsp90β inhibitors and platinum agents such as carboplatin. The results provide a systems-level framework for predicting and overcoming resistance to Hsp90β-targeted therapies.

**Abstract:**

Background/Objectives: Targeting Hsp90β with isoform-selective inhibitors offers a promising therapeutic strategy with reduced toxicity compared to pan-Hsp90 inhibition. However, mechanisms of resistance to Hsp90β-selective inhibition remain poorly defined. This study aimed to identify molecular determinants of Hsp90β dependency and pharmacologic resistance across cancer types. Methods: We integrated gene dependency, transcriptomic, proteomic, metabolomic, and drug sensitivity data from the Cancer Cell Line Encyclopedia with in vitro validation using the Hsp90β-selective inhibitor, NDNB-25. Comparative and correlation analyses were performed to identify resistance-associated pathways, followed by network and combination drug testing to validate functional interactions. Results: Resistant cell lines exhibited extensive rewiring of Rho GTPase signaling, cytoskeletal remodeling, and metabolic adaptation, including mitochondrial dysfunction and redox imbalance. Integrated analyses linked these phenotypes to aryl hydrocarbon receptor (AHR) activation and compensatory Hsp90α expression. Experimental validation confirmed increased kynurenine levels, a known endogenous AHR ligand, in NDNB-25–acquired resistant cells. Gene–drug network integration revealed collateral sensitivity to carboplatin, which synergized with Hsp90β inhibition in resistant models. Conclusions: This study defines the molecular features and adaptive programs underlying resistance to Hsp90β-selective inhibition and identifies therapeutic vulnerabilities that can be exploited to overcome it. The findings establish a systems-level framework for predicting Hsp90β inhibitor response and support rational combination strategies, including carboplatin co-treatment, for future preclinical development.

## 1. Introduction

The 90 kDa heat shock proteins (Hsp90s) are highly conserved molecular chaperones that are critical for cellular homeostasis. These proteins facilitate the proper folding, stabilization, and activation of over 400 client proteins, many of which are involved in essential cellular functions such as signal transduction, protein trafficking, and stress response [1,2,3,4]. In mammalian cells, Hsp90 exists as four isoforms: cytosolic Hsp90α (HSP90AA1) and Hsp90β (HSP90AB1), endoplasmic reticulum-resident Grp94 (HSP90B1), and mitochondrial Trap1 (TRAP1). Each isoform exhibits distinct subcellular localization and client protein specificity. Hsp90α is inducible under stress conditions and supports oncogenic signaling pathways, while Hsp90β is constitutively expressed and plays a key role in cytoskeletal integrity and normal cellular functions. Grp94 primarily assists in the folding of secreted and membrane-bound proteins within the endoplasmic reticulum [5,6], and TRAP1 helps maintain mitochondrial function and protection against oxidative stress [7]. Despite their overlapping functions, these isoforms’ unique roles make them attractive targets for therapeutic intervention in various diseases, including cancer.

Hsp90β, in particular, plays a pivotal role in maintaining cellular homeostasis and supporting essential survival pathways under non-stress conditions. Unlike Hsp90α, which is associated with stress-induced responses, Hsp90β’s constitutive expression makes it indispensable for cell viability and early development [8,9]. Its client proteins include many regulators of cytoskeletal dynamics, cellular adhesion and cell cycle progression [4,10,11,12]. Furthermore, Hsp90β’s involvement in cancer has been increasingly recognized, as it supports the stability and function of numerous oncogenic proteins. Notably, it has been implicated in promoting tumor growth and metastasis through its interactions with proteins involved in these processes. These characteristics underscore its potential as a therapeutic target for the treatment of cancer.

The development of isoform-selective inhibitors has emerged as a promising approach to address the limitations of pan-Hsp90 inhibition [13,14,15], as they exhibit toxicities due to their broad activity against all four Hsp90 isoforms [16,17,18]. Specifically, Hsp90β-selective inhibitors offer a strategy to avoid the on-target effects and pro-survival heat shock response induced by pan-inhibition while preserving therapeutic efficacy. Recent advances in Hsp90β-selective chemistry have yielded next-generation scaffolds with enhanced isoform selectivity and improved pharmacological profiles [13,15,19,20,21], such as novel isoquinolin-1(2H)-based inhibitors characterized for selective client degradation [13,15] and the KUNB106 indazolone derivatives demonstrating potent Hsp90β bias over Hsp90α and mitigated heat shock response activation [19]. In this study, we systematically identified cancer cell populations and molecular pathways that are dependent upon Hsp90β, thereby elucidating the mechanisms of sensitivity and resistance to Hsp90β-selective inhibitors. By integrating multi-omic analyses with high-throughput drug screening datasets, we sought to uncover biomarkers predictive of therapeutic response and explore potential combination strategies to enhance the efficacy of Hsp90β-targeted therapies. Using dependency data, genomics data, and drug screening data from the Cancer Cell Line Encyclopedia (CCLE) in the DepMap database [22,23], this work advances our understanding of Hsp90β biology and provides a foundation for precision oncology approaches toward Hsp90β-dependent cancers.

## 2. Materials and Methods

### 2.1. Cell Line Selection and Classification

Cancer cell lines were classified as HSP90AB1-dependent or HSP90AB1-resistant based on RNAi and CRISPR dependency scores for HSP90AB1, downloaded from the DepMap 24Q2 release [23]. Cell lines were ranked by dependency scores, and representative sensitive and resistant lines were selected from multiple lineages for downstream analyses. NDNB-25 sensitivity data were generated in-house using a panel of 17 DepMap-characterized cell lines, including 22RV1, DBTRG-05MG, HCT116, HepG2, HT29, K562, MCF10A, NCI-H1299, NCI-H23, NCI-H522, OVCAR8, OVCAR5, PANC1, PC3, SH-SY5Y, SKOV3, and T24.

DBTRG-05MG was obtained from the American Type Culture Collection (ATCC, Manassas, VA, USA). All remaining cell lines were obtained as gifts from Dr. Aktar Ali (Biological Screening and Development Core, Warren Family Research Center for Drug Discovery and Development, University of Notre Dame). Most lines were originally sourced from ATCC, with OVCAR8 and OVCAR5 originally obtained from the NCI DCTD Tumor Repository (Frederick, MD, USA).

### 2.2. Cell Culture and Cell Viability Assays

To avoid potential confounding variables in media composition, all cell lines used for this study were grown in DMEM/F-12 (Gibco, Thermo Fisher Scientific, Waltham, MA, USA; Cat. No. 11320033)) containing 1% Penicillin–Streptomycin (VWR International, Radnor, PA, USA; Cat. No. K952-100ML)and 10% Fetal Bovine Serum (FBS Premium; Gibco, Thermo Fisher Scientific, Waltham, MA, USA; Cat. No. A5670701)in an incubator at 37 °C with 5% CO_2_. Cell lines that typically grow in other media formulations were passaged at least 3 times in DMEM/F-12 with 10% FBS.

For cell viability assays, cells were seeded at a density of 5000 cells per well in a 96-well plate for ~24 h before the addition of fresh media and the compound of interest (or vehicle control). Cell viability was measured 72 h after treatment using CellTiter-Glo Luminescent Cell Viability Assay (Promega Corporation, Madison, WI, USA; Cat. No. G9243) according to the manufacturer’s guidelines.

### 2.3. Gene Expression and Proteomic Analysis

Differential gene expression analysis was performed on transcriptomic and proteomic datasets using two-class comparisons or correlation analyses where indicated. Differentially expressed genes (DEGs) were identified using adjusted *p* < 0.05 for transcriptomics and *p* < 0.2 for proteomics datasets, and concordance between transcriptomic and proteomic DEGs was evaluated. We used an adjusted *p* < 0.2 threshold for proteomic analyses to balance discovery sensitivity with the smaller number of quantified proteins and increased technical variability inherent to proteomics datasets. Genes were considered discordant if they had opposing directions (up- vs. downregulated), and discordant DEGs were filtered. Downstream enrichment analyses using concordant transcriptomic/proteomic features were performed in EnrichR.

### 2.4. Metabolomics Analysis

Differential metabolite abundance between Hsp90β-sensitive and -resistant populations was assessed using two-class comparisons or correlation analyses and integrated with gene expression data for joint pathway analysis. Metabolite profiling data were obtained from the Cancer Cell Line Encyclopedia (CCLE) metabolomics compendium available through the DepMap portal, which includes LC–MS quantification normalized to total ion current and batch-corrected across experimental runs using internal standards and pooled reference samples [24]. Normalized metabolite intensities (log_10_-transformed) were used for all downstream analyses in DepMap’s custom analysis toolkit. Pathway-level integration was conducted using enrichment tools compatible with multi-omic inputs, including MetaboAnalyst [25].

### 2.5. LC–MS Analysis of Intracellular Kynurenine in Sensitive and NDNB-25 Acquired Resistant Lines

To measure intracellular kynurenine, we adapted a previously described LC–MS metabolomics extraction and quantification workflow optimized for polar metabolites [26]. Briefly, cells were plated in 10 cm dishes and cultured overnight to ~80% confluency. Monolayers were rapidly washed twice with ice-cold PBS and quenched with 80% methanol (pre-chilled to −80 °C) to extract intracellular metabolites. Cell lysates were scraped, transferred to microcentrifuge tubes, and centrifuged at 14,000× *g* for 10 min at 4 °C. Supernatants were collected and evaporated to dryness using a SpeedVac Concentrator (Thermo Fisher Scientific, Waltham, MA, USA). Dried extracts were resuspended in 90% acetonitrile/10% water and analyzed by LC–MS using an Agilent 1290 Infinity II UHPLC System (Agilent Technologies, Santa Clara, CA, USA) coupled to an Agilent 6460 Triple Quadrupole Mass Spectrometer (Agilent Technologies, Santa Clara, CA, USA) using Masshunter Software Suite (B.09.00) operating in positive ion mode.

Chromatographic separation was performed using a hydrophilic interaction liquid chromatography (HILIC) method with a gradient of acetonitrile and ammonium acetate buffer, as previously described [26]. For kynurenine quantification, a pure kynurenine reference standard (Sigma-Aldrich, Merck KGaA, Darmstadt, Germany; Cat. No. K8625) was used to establish retention time and confirm ion transitions using two multiple-reaction monitoring (MRM) pairs. A standard curve generated from serial dilutions of pure kynurenine was used to verify assay linearity. Because the primary objective was to compare relative intracellular kynurenine abundance across cell lines, abundance was normalized to the signal intensity of the control (parental cell line) rather than absolute quantification.

### 2.6. Gene Dependency and Functional Enrichment Analysis

CRISPR gene effect data from DepMap were used in two-class comparisons or correlation analyses to generate average dependency profiles for sensitive and resistant groups. Dependencies were filtered by significance (*p* < 0.05) and functional relevance (score < −0.3 and intergroup difference > 0.2) to identify context-specific vulnerabilities. Enrichment of pathway-level dependencies was assessed using the gene set enrichment tool EnrichR (Ma’ayan Lab, Icahn School of Medicine at Mount Sinai) [27].

### 2.7. Drug Sensitivity Analysis

Drug response data were obtained from the PRISM Repurposing (24Q2) dataset and Genomics of Drug Sensitivity in Cancer (GDSC) [23]. Compounds differentially active between sensitive and resistant lines were identified using two-class comparisons and correlation analyses. A viability filter (≥50% reduction in cell viability) was applied to identify active compounds for enrichment analysis.

### 2.8. Integrated Network and Pathway Analysis

Drug–target associations were derived from DGIdb (v5.0.9) and DrugEnrichR (Ma’ayan Lab, Icahn School of Medicine at Mount Sinai) [28,29]. Shared and unique features across omics platforms, gene dependencies, and drug targets were visualized using Venn diagrams, and network models were built using STRING protein–protein interactions (PPIs) [30] (Version 12.0) and visualized in Cytoscape (V.3.10.2) [31]. Integrated enrichment analyses were also performed using PANGEA (Version 1.1) to identify convergent resistance pathways [32].

### 2.9. NDNB-25 Resistance Modeling and Combination Drug Testing

NDNB-25-resistant lines were generated by serially passaging NDNB-25-sensitive NCI-H23 cells in increasing concentrations of NDNB-25. Single-drug and combination synergy screening was conducted by treating sensitive and resistant lines with varying concentrations of NDNB-25 and carboplatin. Cell viability was measured 72 h after treatment using CellTiter-Glo. Synergy scores were computed using SynergyFinder (Version 3) with the ZIP synergy model [33].

### 2.10. Western Blot Analysis

Approximately 300,000 cells were seeded into 6-well plates and grown to ~80% confluency. The media was replaced with fresh media containing NDNB-25 at the indicated concentrations or vehicle control for 24 h. Cells were lysed using RIPA buffer containing protease and phosphatase inhibitors, and protein concentration in lysates was determined by Pierce BCA Protein Assay Kit (Thermo Fisher Scientific, Waltham, MA, USA; Cat. No. 23227). Proteins were separated by SDS-PAGE using 10% SDS-polyacrylamide gels and transferred to nitrocellulose membranes. Transfer efficiency and total protein were evaluated by Ponceau staining. Membranes were blocked with 5% nonfat dry milk solution and probed with antibodies specific to GAPDH (Cell Signaling Technology, Danvers, MA, USA; Cat. No. 2118), Hsp90α (Enzo Life Sciences, Farmingdale, NY, USA; Cat. No. ADI-SPA-840), CDK4 (Cell Signaling Technology, Danvers, MA, USA; Cat. No. 12790), and Akt (Cell Signaling Technology, Danvers, MA, USA; Cat. No. 9272). Antibody dilutions were 1:1000, and secondary antibodies were used at 1:5000 (Southern Biotech, Birmingham, AL, USA; Rabbit Cat. No. 4030-05, Rat Cat. No. 3030-05 ). Blots were developed using Clarity Western ECL Substrate (Bio-Rad Laboratories, Hercules, CA, USA; Cat. No. 1705060) on a ChemiDoc Imaging System (Bio-Rad Laboratories, Hercules, CA, USA).

## 3. Results

### 3.1. Identification and Characterization of HSP90AB1-Dependent and -Resistant Cancer Cell Lines

CRISPR dependency data for the HSP90AB1 gene (Hsp90β) from the DepMap portal was used to identify putative Hsp90β-sensitive and -resistant populations. A marginal lineage bias was observed, with ovarian and cervical cancers among the most resistant, while bone and myeloid cell lines were among the most dependent (Figure 1A). HSP90AB1-dependent and -resistant cell lines from multiple lineages, which were identified based on both CRISPR and RNAi dependency scores with emphasis on those that exhibit the strongest dependency or resistance across both datasets, were selected for preliminary analyses (Figure 1B, Appendix A).

A custom workflow was developed to compare omics datasets, gene dependencies, and drug sensitivities using two-class comparisons between HSP90AB1-dependent and -resistant populations (Figure 1C). This approach enabled the identification of overlapping genetic alterations driving functional differences between groups. Functional enrichment and integrated pathway analyses were conducted on differentially expressed genes (DEGs), metabolites, over-represented damaging mutations, and gene dependencies to identify key pathways of interest. An example of a two-class comparison using damaging mutations and a downstream enrichment analysis is shown in Appendix A and Appendix A.

### 3.2. Gene Expression and Metabolite Profiling Reveal Distinct Metabolic and Signaling Programs in HSP90AB1-Dependent and -Resistant Cells

Pathways associated with resistance to HSP90AB1 genetic ablation (CRISPR and RNAi) were identified using DEGs derived from transcriptomic and proteomic two-class comparisons between dependent and resistant cells. Strong concordance was observed between transcriptomic and proteomic DEGs (Appendix A, Appendix A). Functional enrichment analysis highlighted broad differences between the groups, particularly involving signaling pathways and structural elements (Figure 2A,B, Appendix A).

Metabolic alterations that contribute to resistance were further explored through a two-class comparison of metabolite abundance between dependent and resistant cells (Appendix A). Integration of DEG and metabolomics data in a joint pathway analysis (Figure 2C) revealed significant dysregulation of metabolic pathways, including pyrimidine metabolism, purine metabolism, and arginine biosynthesis (Figure 2D and Appendix AB,C).

In addition to metabolic differences, several key signaling pathways and structural features were identified as distinguishing between groups. Rap1, PI3K-AKT, and Hippo signaling pathways were differentially regulated, as were structural components including tight junctions and actin cytoskeleton organization (Figure 2E,F and Appendix AD–F). These findings support a model in which resistance to HSP90AB1 genetic ablation is driven by both metabolic reprogramming and alterations in signaling and cellular architecture.

### 3.3. Integrated Gene Dependency and Drug Sensitivity Profiling Identifies Therapeutic Vulnerabilities in HSP90AB1-Resistant Cells

An integrative strategy was used to identify therapeutic vulnerabilities in HSP90AB1-resistant cells. Gene dependency profiles were generated for dependent and resistant lines using DepMap CRISPR gene effect data (Appendix A). Enrichment analysis identified that multiple genes that facilitate RNA processing and Rho GTPase signaling are indispensable features of resistant cells (Figure 3A), while dependent cells exhibited co-dependencies related to Hsp90 function and stress response pathways (Figure 3B).

Complementary to the gene dependency analysis, drug sensitivity profiles were generated using the PRISM Repurposing 24Q2 dataset (Appendix A, Appendix A) and IC_50_ values from the Genomics of Drug Sensitivity in Cancer Project (Figure 3C, Appendix A). DrugEnrichR was used to identify common gene targets of compounds that selectively targeted either HSP90AB1-dependent or -resistant populations. (Appendix A).

To contextualize these findings, shared and unique genes were identified across -omics comparisons, gene dependencies, and drug target datasets. Although direct gene-level overlap among these datasets was limited (Figure 3D and Appendix AB), pathway-level relationships were explored using the PANGEA enrichment tool. This approach highlighted distinct therapeutic vulnerabilities in resistant lines, particularly within cytoskeletal organization and motility (Rho GTPase), oncogenic signaling (Ras/MAPK/ERK), and stem cell maintenance (WNT/β-catenin) (Figure 3E). In contrast, HSP90AB1-dependent lines were enriched for vulnerabilities in DNA damage repair and response pathways (Appendix A).

Rho GTPase dependency in the resistant cells showed overlap with mutations and differential gene expression, suggesting a potential role in the mediation of resistance. Network analyses incorporating Hsp90β interactors, gene sets, and compounds selectively targeting resistant lines were performed to investigate this further. Integration of protein–protein and drug–protein interactions revealed Hsp90β interactors that may contribute to resistance and identified key nodes underlying specific dependencies (Figure 3F).

### 3.4. Shared and Unique Characteristics of Cell Lines Resistant to Hsp90β-Specific Inhibitor NDNB-25

Validation of the multi-omics gene dependency analysis and model for HSP90AB1 resistance was performed by testing 17 cell lines from the DepMap database using the Hsp90β-selective inhibitor NDNB-25. As anticipated, these lines exhibited a broad range of NDNB-25 IC_50_ values (Figure 4A, Appendix A). Notably, IC_50S_ for NDNB-25 were more strongly correlated with HSP90AB1 RNAi dependency than CRISPR-based dependency (Figure 4B). Additional confirmation was provided by evaluating a second Hsp90β-selective inhibitor, NDNB-21, in a subset of NDNB-25-sensitive and -resistant lines, which exhibited a similar differential IC_50_ response (Appendix A).

The next objective was to identify features associated with both HSP90AB1 dependency and NDNB-25 sensitivity, as well as mechanisms of NDNB-25 resistance that may function independently of, or upstream from, the inhibition of Hsp90β. Damaging mutations were compared between NDNB-25-sensitive and -resistant cell lines, revealing distinct mutation profiles enriched within each group (Appendix A). Among sensitive lines, TTN—encoding the structural protein titin—was the most frequently mutated gene (Figure 4C). However, the limited overlap of specific mutations within each group suggested that divergent mutations might converge on common biological pathways. Enrichment analyses of these mutations uncovered shared processes that were more evident in sensitive than in resistant lines (Figure 4D), particularly in pathways related to metabolism and stress response.

Subsequent multi-omics correlation analyses, structured similarly to those in Figure 2 and Figure 3, revealed that the lower NDNB-25 IC_50S_ in sensitive lines were associated with the upregulation of genes involved in mitochondrial function as well as metabolism. In contrast, resistant lines showed an enrichment in pathways associated with Rho GTPase signaling, homeostatic regulation, and stress adaptation (Appendix A, Appendix A). Notably, HSP90AA1, the gene encoding Hsp90α, was basally higher in NDNB-25-resistant lines.

The overlap of proteomic correlations between HSP90AB1 RNAi and NDNB-25 IC_50_ analyses was assessed to identify features consistent between groups (Appendix A). A total of 66 differentially expressed genes were shared between both datasets (Figure 4E), representing the most robust expression markers of Hsp90β dependency. These shared DEGs reinforce the importance of key signaling and regulatory pathways (including Rho GTPase signaling, cytoskeletal remodeling, RNA processing, receptor tyrosine kinase (RTK) signaling, and phosphoinositide metabolism) that contribute to the cellular response to Hsp90β inhibition, regardless of whether inhibition occurs through genetic silencing or pharmacologic targeting (Figure 4F).

To validate some of these findings, we generated an NDNB-25-resistant line through serially passaging the NDNB-25-sensitive line NCI-H23 at increasing concentrations of NDNB-25 (Appendix A). Importantly, the NDNB-25-acquired resistant line also had basally high HSP90α expression, and there was no degradation to Hsp90 client proteins in response to NDNB-25 in the resistant line (Figure 4G and Appendix A), validating that compensatory Hsp90α expression and function are a feature of resistance.

Analysis of DepMap/CCLE metabolomics data revealed a feature unique to NDNB-25; kynurenine exhibited the strongest positive correlation with NDNB-25 resistance, whereas its precursor tryptophan was negatively correlated (Figure 4H and Appendix A). These associations were not observed in the earlier analysis for the HSP90AB1 gene dependency analyses shown in Figure 2. Kynurenine is an endogenous ligand for the aryl hydrocarbon receptor (AHR)—a transcriptional regulator of xenobiotic metabolism, redox balance, and mitochondrial function—suggesting a mechanistic link between AHR signaling and NDNB-25 resistance. Supporting this hypothesis, integrated gene–metabolite joint pathway analysis highlighted mitochondrial and energy homeostasis pathways as significantly altered in NDNB-25-resistant lines (Appendix A; Appendix A). Notably, several top-correlated features (including PDP1, NDUFA5, COX5B, ALDH1L1, and MTHFD2, as well as NAD, pyroglutamic acid, methionine sulfoxide, and thiamine levels) are downstream of or functionally linked to AHR activity and suggest mitochondrial dysfunction and redox remodeling as contributors to resistance. LC–MS analysis confirmed kynurenine as a distinguishing metabolic feature of NDNB-25 resistance, with significantly elevated intracellular levels in the NDNB-25-acquired resistant NCI-H23 line compared with its parental counterpart (Figure 4I).

### 3.5. Comparative Analysis of HSP90AB1 Gene Dependency and NDNB-25 Sensitivity Reveals Unique Co-Dependencies and Drug Sensitivities

Correlation analyses were performed between NDNB-25 IC_50_ values and RNAi gene dependencies from the DepMap dataset to identify co-dependencies in NDNB-25-sensitive lines and potential therapeutic targets in resistant lines (Appendix A). These NDNB-25-specific co-dependencies were compared with those associated with HSP90AB1 dependency to distinguish shared resistance mechanisms from those unique to NDNB-25 (Appendix A). Gene dependencies significantly correlated with NDNB-25 IC_50_ values (*p* < 0.05) and were further evaluated for differences between NDNB-25-resistant and HSP90AB1-resistant lines (Figure 5A).

An iterative filtering strategy was employed to identify subtle but meaningful differences in dependency profiles. A dependency score threshold of < −0.3 was applied to capture genes with moderate to strong essentiality. While a score near −1 is typically indicative of common essential genes, a threshold of −0.3 was chosen to capture context-specific vulnerabilities that may not be universally essential but are still functionally relevant in resistant populations. Additionally, a minimum difference of >0.2 between groups was required to ensure biological relevance. This approach identified six genes with higher dependency in NDNB-25-resistant lines (Figure 5B). A similar analysis was applied to gene expression data to identify group-specific transcriptomic differences (Appendix A).

Enrichment analyses were performed on both differentially expressed genes and dependencies to gain mechanistic insight. This analysis revealed pathways that are unique to NDNB-25 resistance, including metabolic processes, drug detoxification, stress response pathways, nervous system development, and Golgi-associated trafficking and transport (Appendix A).

Gene dependency and PRISM Repurposing drug sensitivity data were also analyzed in NDNB-25-sensitive and -resistant populations to support integrative downstream analyses (Figure 5C,D). PRISM compounds were first filtered based on their correlation with NDNB-25 IC_50_ values (*p* < 0.05), and representative examples of compounds showing distinct positive or negative correlations are shown (Appendix A). A viability-based filter was then applied, selecting only compounds that decreased cell viability by at least 50% in either group (Appendix A). This yielded approximately 40 compounds that selectively targeted NDNB-25-sensitive or -resistant populations (Figure 5E).

### 3.6. Integrated Gene–Drug Network Analysis Informs Rational Combination Screening to Validate Mechanisms of NDNB-25 Resistance

An integrated interaction network was constructed to contextualize the gene dependencies, multi-omics signatures, and drug sensitivities identified in earlier analyses (Figure 6A). This network incorporates differentially expressed genes, gene dependencies, damaging mutations, and compounds identified from the PRISM Repurposing screen. Hsp90β interactors and gene–drug interactions curated from DGIdb were also included to bridge unknown connections between drug sensitivities and genomic features. Enrichment analysis of the NDNB-25-resistant network (Figure 6B) identified several key signaling and structural pathways for further exploration.

Multiple approaches were used to explore the connection of key nodes within a pathway of interest. An example of a subset related to regulation of the actin cytoskeleton, within the broader NDNB-25-resistant network, is demonstrated in Appendix A. The pathway term was used to subset relevant genes within the network (Appendix A). The largest subset of interconnected nodes was then extracted to form a focused subnetwork (Appendix A).

Signaling in Rho GTPases was interrogated with this approach and yielded a highly interconnected module (Figure 6C). This subnetwork contains multiple DEGs elevated in resistant lines (NRP1, S100A2, VCL, ANXA2, ANXA6, SORBS3), as well as genes that are both differentially expressed and known interactors of HSP90β [34] (CD151, EZR, RAB8A). Several key Hsp90β interactors and drug targets, including EGFR, FGFR1, ABL1, TP53, and FGR, also reside within this network. The structure and connectivity of this subnetwork highlight a convergence of NDNB-25 resistance mechanisms on cytoskeletal remodeling and signaling pathways and reveal key nodes connected to carboplatin sensitivity. The network analysis was further explored using a subnetwork of Hsp90β interactors, which revealed a large network of HSP90AA1 interactors that are upregulated in resistant lines, indicating at least some compensation by basally high Hsp90α expression (Appendix A).

Carboplatin was one of the top hits from our drug sensitivity analysis in NDNB-25-resistant lines (see Appendix A); therefore, NDNB-25-sensitive and -resistant lines were treated with carboplatin in combination with NDNB-25 at varying drug concentrations. In support of our model, there was a strong synergistic effect in the NDNB-25-resistant lines and not the NDNB-25-sensitive lines (Figure 6D–G). We also tested the NDNB-25-acquired resistant line, and carboplatin had a higher synergy score with NDNB-25 in this context (Figure 6H,I). These findings support a model in which NDNB-25-resistant cells may engage stress-adaptive, pro-motility programs that are amenable to therapeutic targeting through combination strategies.

## 4. Discussion

Recent development of isoform-selective Hsp90β inhibitors has renewed interest in understanding the cellular consequences and resistance mechanisms associated with HSP90AB1 inhibition [13,15,19]. This study broadly defines the molecular landscape of resistance to HSP90AB1 gene dependency and Hsp90β-selective inhibition through a comprehensive integration of gene expression, metabolomic, co-dependencies and drug sensitivity data. By delineating distinct cellular states associated with HSP90AB1 dependency and resistance to the Hsp90β-selective inhibitor NDNB-25, we identified convergent resistance programs centered on cytoskeletal remodeling, metabolic adaptation, and the stress response.

We implemented a flexible multi-omic analytical framework that balanced hypothesis-driven comparisons with data-guided exploration to dissect these phenotypes. Comparisons between defined sensitive and resistant populations were used for many analyses, including differential gene expression and metabolite abundance, enabling robust pathway enrichment and joint pathway modeling (Figure 2 and Figure 3). In parallel, correlation analyses were employed where continuous relationships, such as drug response metrics or dependency scores, offered complementary insights. While statistical thresholds were kept deliberately permissive during exploratory stages to avoid overlooking subtle trends, most final analyses applied standard significance cutoffs. Rather than adhering to a rigid pipeline, we prioritized interpretability and biological coherence, allowing the structure of the data to guide the selection of analytical methods. This adaptive framework is aligned with recent large-scale exploratory efforts leveraging DepMap and CCLE resources to identify context-specific dependencies and biomarkers of drug sensitivity [35,36,37,38,39].

Analyses reveal that resistance to Hsp90β inhibition is associated with rewiring of Rho GTPase signaling and cytoskeletal integrity pathways, implicating regulators such as IQGAP1, ARHGDIA, VCL, and ANXA family members. These proteins coordinate actin dynamics, vesicle trafficking, and focal adhesion maintenance—functions critical for preserving cellular architecture under proteotoxic stress. Resistant cells appear to circumvent Hsp90β dependency by engaging scaffolding networks and adaptive signaling modules that stabilize the cytoskeleton, thereby maintaining essential survival signals. Similar cytoskeletal remodeling has been implicated in pan-Hsp90 dependency, including signaling related to AKT and adhesion molecules that stabilize cellular architecture under stress [40].

Metabolic rewiring emerges as a hallmark of resistance with both shared and unique pathways of interest between HSP90AB1 dependency and Hsp90β inhibition. Integrated transcriptomic–metabolomic correlation analyses implicate altered NAD+ metabolism, mitochondrial redox balance, and one-carbon cycle intermediates (e.g., MTHFD2, ALDH1L1, methionine sulfoxide) in NDNB-25 resistance. Notably, increased kynurenine levels and a corresponding depletion of tryptophan highlight upregulation of the kynurenine pathway. This axis is not only critical for NAD+ biosynthesis but also activates the aryl hydrocarbon receptor (AHR), a ligand-activated transcription factor that governs cellular responses to xenobiotics and oxidative stress.

AHR activation has been linked to multidrug resistance, modulation of Rho GTPases, and regulation of drug efflux transporters, potentially contributing to NDNB-25 resistance through both transcriptional and post-translational mechanisms [41,42,43,44,45]. Importantly, AHR is itself a canonical client of Hsp90 in the unliganded, inactive state, with the Hsp90–AHR complex maintaining AHR stability and ligand responsiveness [46,47,48]. Upon binding of kynurenine (an endogenous AHR ligand derived from tryptophan metabolism), AHR dissociates from Hsp90 and translocates to the nucleus to regulate downstream targets. Thus, elevated kynurenine levels in NDNB-25-resistant cells may promote AHR activation via this established chaperone-dependent mechanism. The compensatory upregulation of Hsp90α observed in our resistant lines could further modulate AHR dynamics by shifting client competition or altering chaperone availability. To experimentally validate this metabolic signature, we quantified intracellular kynurenine by LC–MS in paired NDNB-25–sensitive and NDNB-25–acquired resistant NCI-H23 lines. Kynurenine levels were significantly elevated in resistant cells, consistent with activation of the kynurenine–AHR axis (Figure 4I). These results provide direct biochemical evidence supporting the metabolomic and transcriptomic correlations. Ongoing studies are evaluating the functional relevance of AHR activation using pharmacologic and genetic inhibition approaches.

A key insight from this study is the partial divergence between mechanisms underlying genetic HSP90AB1 dependency and pharmacologic resistance to NDNB-25. While both phenotypes converge on cytoskeletal and metabolic programs, NDNB-25-resistant cells show unique vulnerabilities related to drug detoxification, Golgi trafficking, and vesicular transport. This distinction was supported by co-dependency analysis and drug screening, which identified selective vulnerabilities in NDNB-25-resistant lines that were not observed in HSP90AB1-resistant cells. These distinctions underscore the need to evaluate pharmacologic resistance through multi-layered profiling, rather than relying solely on genetic dependency data, and justify the pathway-centric approach taken in this study.

Our integrative network analysis revealed that resistance converges on interconnected modules involving Rho GTPases, PI3K/AKT signaling, and receptor tyrosine kinase hubs, including EGFR and FGFR1—known interactors of Hsp90β. Importantly, our analyses revealed a connection to increased sensitivity to carboplatin, a platinum-based chemotherapeutic agent. Similar integrative network strategies have been employed to identify convergent vulnerabilities and drug repurposing opportunities in resistance settings [37,49,50]. Functional validation experiments demonstrated a robust synergy between carboplatin and NDNB-25 in resistant lines, but not in sensitive lines, supporting a model in which stress-adaptive resistance programs confer collateral sensitivity to DNA-damaging agents. These findings suggest that combining Hsp90β-selective inhibitors with platinum agents may be an effective strategy to overcome drug resistance. Basally high Hsp90α expression was a feature present in resistant lines. Moreover, resistant lines had higher expression of some, but not all, Hsp90α interactors. Consistent with our findings, a recent study in BCR-ABL1+ leukemia demonstrated that loss of Hsp90β function induces compensatory Hsp90α upregulation, highlighting the non-redundant and context-specific roles played by each isoform in resistance [51]. Furthermore, their work further supports the therapeutic value of isoform-selective inhibitors and rational combination strategies targeting adaptive stress responses.

Limitations to this study include the primarily correlative nature of the analyses and the limited scope of experimental validation. While our multi-omic framework was designed to provide a broad and integrative characterization of Hsp90β resistance across diverse cancer contexts, more rigorous mechanistic validation will be required to confirm the causality of specific pathways identified here. The present work was intended as a discovery-oriented effort to uncover convergent programs of resistance and prioritize candidate mechanisms, such as compensatory Hsp90α upregulation and kynurenine–AHR signaling, for targeted investigation. Ongoing and future studies will employ lineage-specific models and context-dependent perturbation experiments to dissect these mechanisms in greater depth.

Our decision to emphasize pathway-level and gene program–level features, rather than individual genes, was guided by the complexity and heterogeneity of Hsp90β biology across cancer types. Pathway-level changes provide more robust and generalizable biomarkers of resistance that are less sensitive to experimental noise and context-specific variability. Future investigations will integrate additional clinically relevant data types, including copy number variation (CNV), DNA methylation, and gene fusions, to refine predictive models of Hsp90β dependency and resistance. Correlating these molecular features with drug sensitivity, such as carboplatin synergy, and with clinical outcomes in patient-derived datasets may help identify diagnostic markers for treatment stratification and therapeutic response prediction.

Although NDNB-25 and NDNB-21 have not yet been evaluated clinically, earlier Hsp90β-selective analogs, such as NDNB1182, demonstrated in vivo efficacy and favorable safety profiles when combined with immune checkpoint blockade in murine tumor models [52]. These findings indicate that selective Hsp90β inhibition can achieve pharmacologically relevant exposures with minimal toxicity. Our current lead compounds are direct descendants of these scaffolds and are presently undergoing preclinical pharmacokinetic, pharmacodynamic, and toxicology evaluation to support in vivo testing and translational development.

Finally, the analytical framework presented here can be broadly applied to other chaperone systems and stress-adaptive pathways. By integrating multi-layered omics data with experimental validation, future studies can extend these findings to delineate Hsp90β’s roles not only in cancer resistance but also in normal cellular biology and tissue-specific proteostasis, an important and understudied aspect of Hsp90β function in the context of Hsp90 isoform-selective inhibitors.

## 5. Conclusions

This study establishes a comprehensive multi-omic framework for understanding resistance to Hsp90β-selective inhibition. Resistance involves convergent adaptations in Rho GTPase signaling, cytoskeletal organization, and metabolic pathways, notably activation of the kynurenine–AHR axis and compensatory Hsp90α upregulation. Integrative network analyses identified collateral sensitivity to carboplatin, which synergized with NDNB-25 in resistant models. These findings highlight both the therapeutic potential and complexity of isoform-specific Hsp90 targeting, providing a foundation for rational drug combination strategies. Future studies integrating lineage-specific models, additional omics data, and patient-derived samples will refine these predictive biomarkers and support translational development of Hsp90β inhibitors.

## Figures and Tables

**Figure 1 cancers-17-03488-f001:**
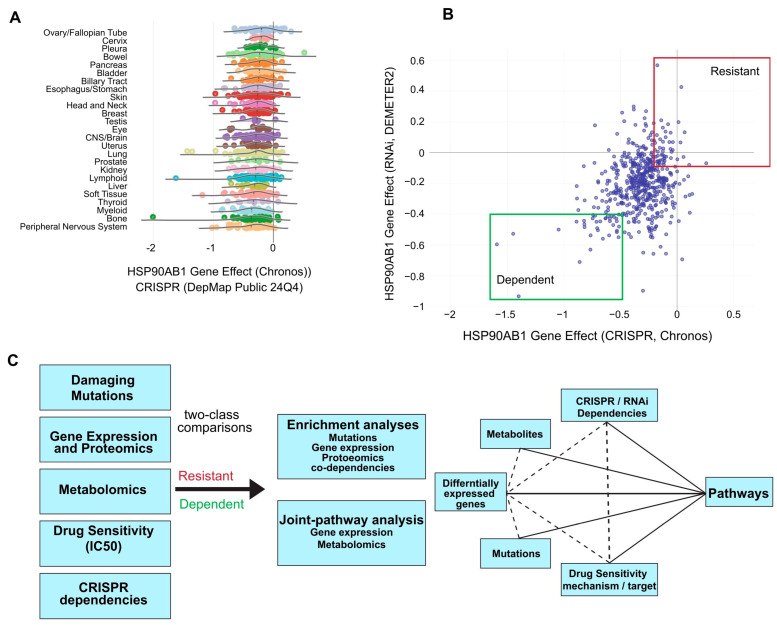
Identification of HSP90AB1-dependent and -resistant cancer cell populations. (**A**) Distribution of HSP90AB1 CRISPR dependency scores across cancer cell lines in DepMap, stratified by cancer lineage. (**B**) Classification of putative cell lines as sensitive (green box) or resistant (red box) based on HSP90AB1 CRISPR and RNAi dependency data. Selected cell lines used in downstream analyses are outlined. (**C**) Overview of the comparative analysis workflow, integrating omics data, gene dependencies, and drug sensitivities between sensitive and resistant groups.

**Figure 2 cancers-17-03488-f002:**
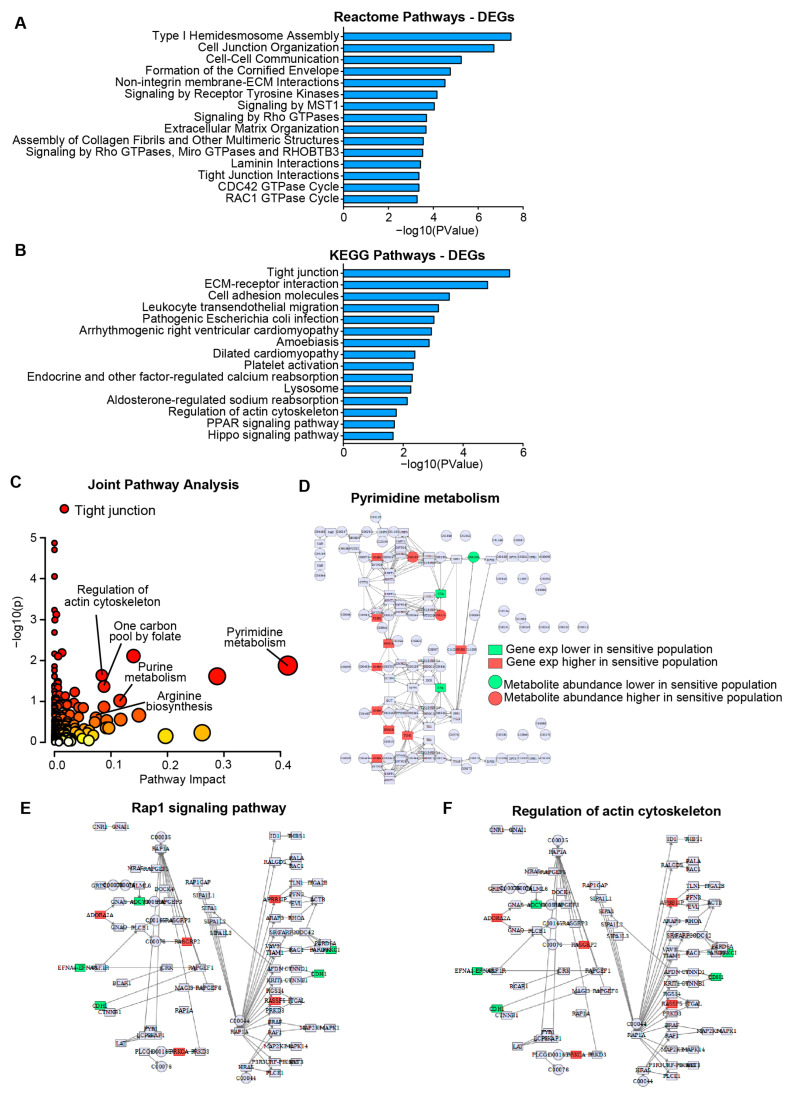
Gene expression and metabolite profiling reveal distinct metabolic and signaling programs in HSP90AB1-dependent and -resistant cells. (**A**,**B**) Enrichment analysis of differentially expressed genes (DEGs) showing Reactome (**A**) and KEGG (**B**) pathways significantly altered between dependent and resistant cell lines. (**C**) Joint pathway analysis integrating DEGs and differentially abundant metabolites. Individual pathways scored by significance and pathway impact with point sizes scaled to pathway impact (x axis) and color scaled to significance (y axis). Kegg Pathview maps shown to illustrate broad differences. See Appendix A for individual genes/metabolites. Significantly altered genes and metabolites in dependent and resistant populations related to pyrimidine metabolism (**D**) and DEGs in signaling and structural pathways including Rap1, and actin cytoskeleton organization (**E**,**F**).

**Figure 3 cancers-17-03488-f003:**
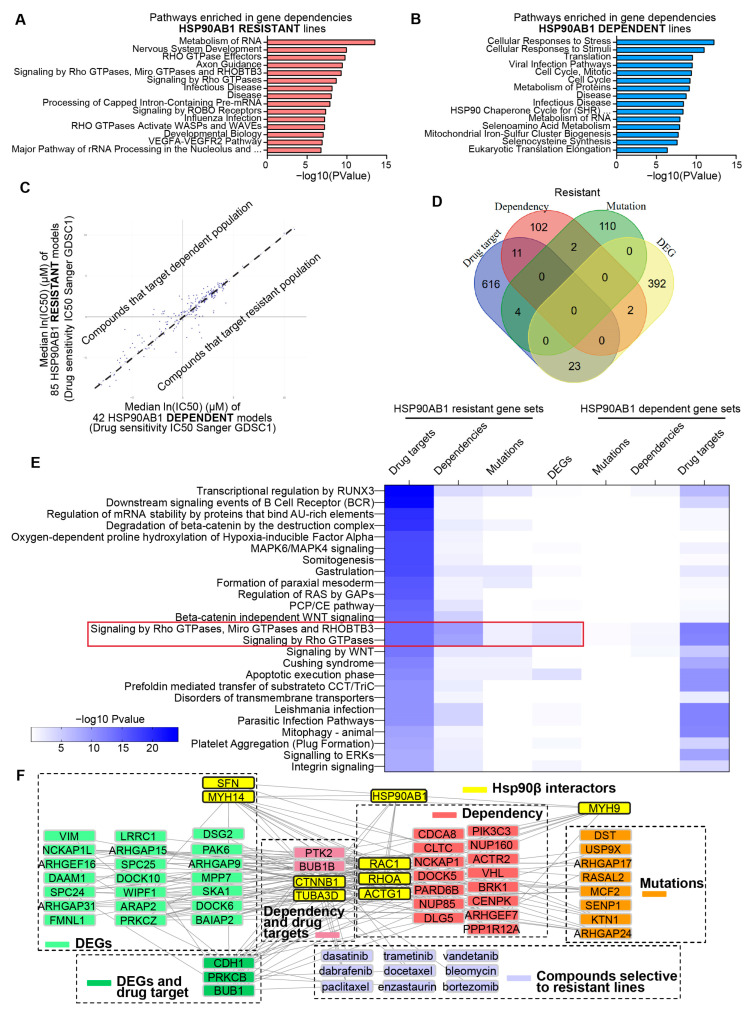
Integrated gene dependency and drug sensitivity profiling identifies therapeutic vulnerabilities in HSP90AB1-resistant cells. (**A**,**B**) Enriched pathways among CRISPR dependencies in resistant lines (**A**) and sensitive lines (**B**). (**C**) Differential drug sensitivities from The Genomics of Drug Sensitivity in Cancer (GDSC) datasets across sensitive and resistant populations visualized by X/Y plot with median IC50 for sensitive lines on the x-axis and resistant lines on the y-axis. (**D**) Overlap of gene dependencies, mutations, DEGs, and drug targets in resistant lines. (**E**) Pathway enrichment from multiple enrichment analyses (PANGEA), highlighting the top 25 pathways for the genes that are drug targets in resistant lines compared to parallel analyses in DEGs, dependencies and drug target gene lists. (**F**) Network analysis for visualizing protein–protein interactions (PPIs), drug–protein interactions and known Hsp90β interactors contributing to resistance.

**Figure 4 cancers-17-03488-f004:**
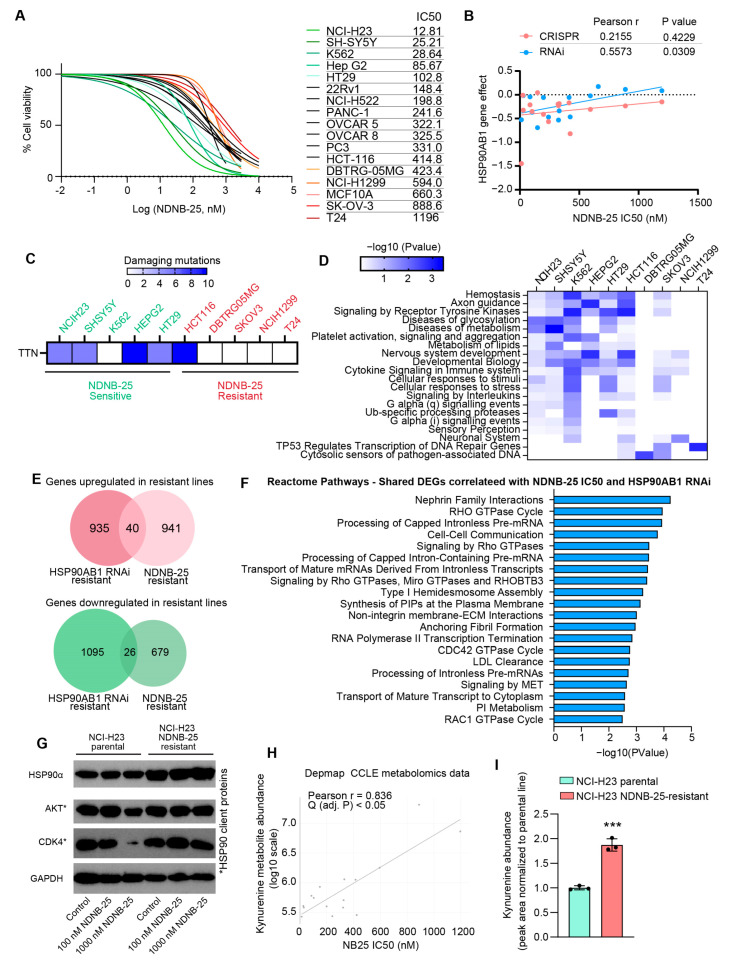
Shared and unique molecular features of NDNB-25-sensitive and -resistant populations. (**A**) NDNB-25 IC_50_ values across tested cell lines that have omics and dependency data available in DepMap. (**B**) Correlation between NDNB-25 IC_50_ and HSP90AB1 RNAi and CRISPR dependency. (**C**) Most frequent damaging mutations enriched in NDNB-25-sensitive lines. (**D**) Pathway enrichment of damaging mutations in sensitive and resistant lines. (**E**) Shared DEGs between NDNB-25 IC_50_ and HSP90AB1 RNAi correlation analyses. (**F**) Enriched pathways among shared DEGs, including Rho GTPase signaling, cytoskeletal remodeling, RTK signaling, and phosphoinositide metabolism. (**G**) Western blot analysis of Hsp90α and Hsp90 client proteins in NCI-H23 parental and NDNB-25-acquired resistant cells generated by serial passaging in escalating NDNB-25 concentrations. (**H**) X/Y plot of Pearson correlation analysis between NDNB-25 sensitivity (IC50) and kynurenine abundance from the Depmap/CCLE metabolomics dataset. (**I**) LC–MS quantification of intracellular kynurenine in paired NCI-H23 parental and NDNB-25-acquired resistant cells. Kynurenine measurement by multiple-reaction monitoring (MRM) peak areas using the transitions *m*/*z* 209.1 → 94.1 (qualifier) and *m*/*z* 209.1 → 192.0 (quantifier). Data were analyzed by a two-tailed Student’s *t*-test, *p* < 0.001 (***). The uncropped blots are shown in Appendix A.

**Figure 5 cancers-17-03488-f005:**
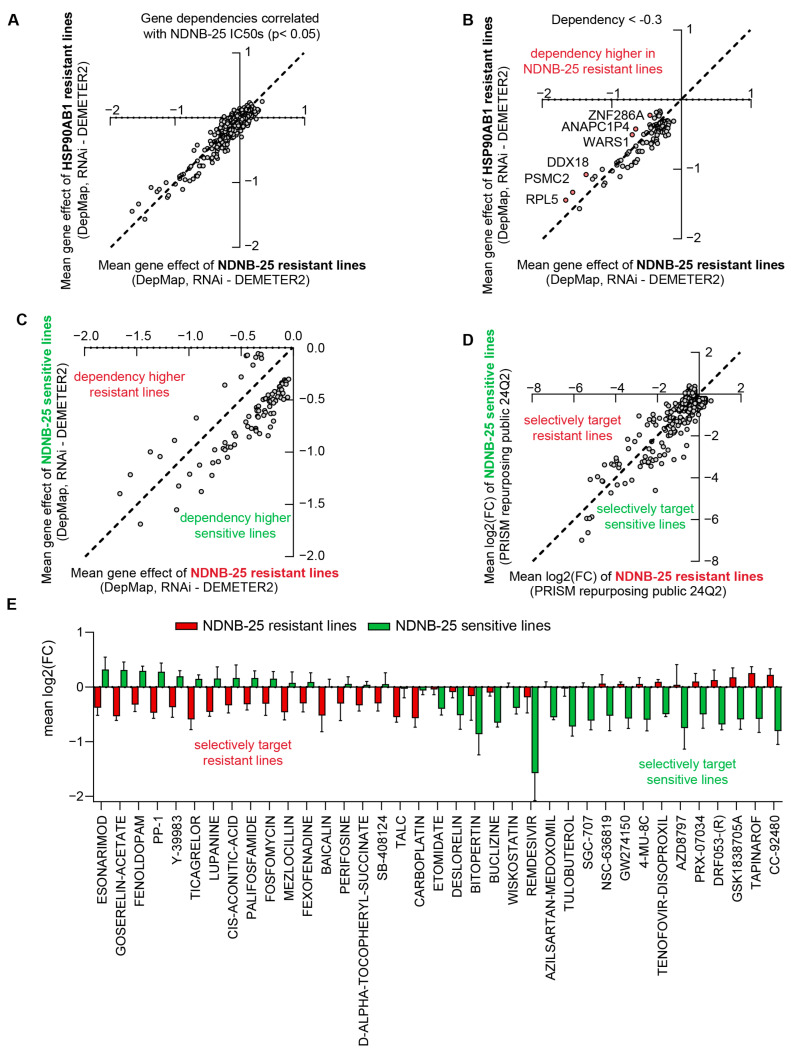
Comparative analysis of HSP90AB1 gene dependency and NDNB-25 sensitivity reveals unique co-dependencies and drug sensitivities. (**A**,**B**) Gene dependencies (RNAi) were correlated with NDNB-25 IC_50_ values (*p* < 0.05), and X/Y plots were generated to show dependency differences (group mean for NDNB-25-resistant or HSP90AB1-resistant lines) comparing NDNB-25-associated dependencies and HSP90AB1 co-dependencies (**A**). Genes with higher dependency in NDNB-25-resistant lines were identified using a dependency score threshold (<−0.3) and intergroup difference (>0.2). (**C**,**D**) X/Y plots comparing gene dependency (**C**) and drug sensitivity (**D**) profiles for NDNB-25-sensitive and -resistant lines. (**E**) Compounds that selectively target sensitive or resistant lines (from (**D**)) filtered by compounds that cause ≥ 50% reduction in cell viability in either population.

**Figure 6 cancers-17-03488-f006:**
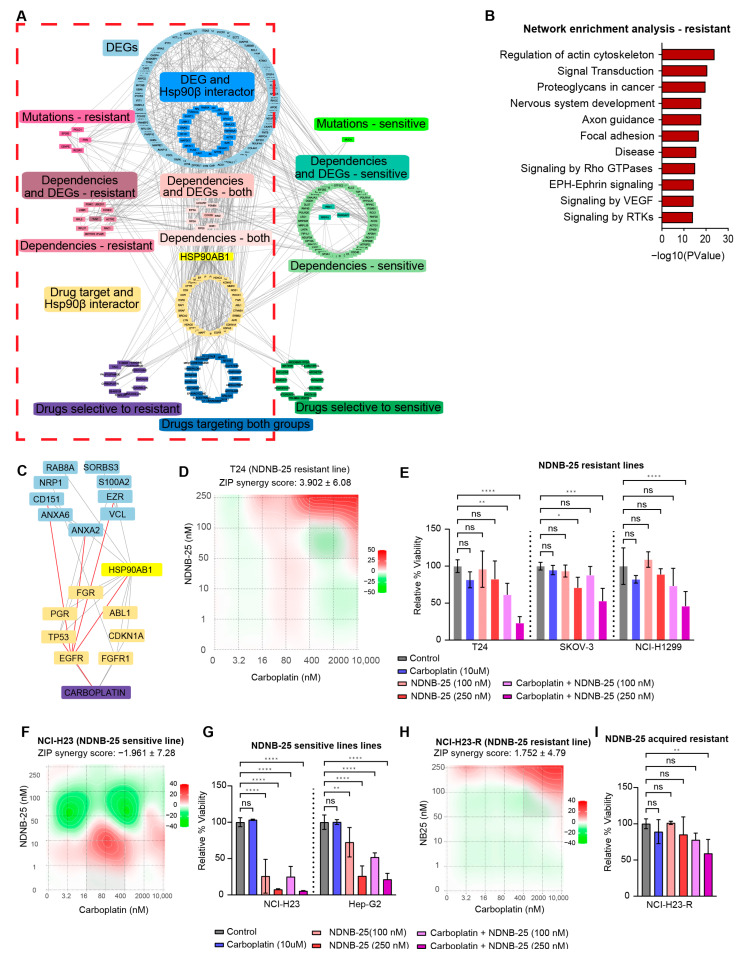
Integrated network analysis and functional validation of therapeutic vulnerabilities in NDNB-25-resistant cells. (**A**) Gene–drug-dependency interaction network incorporating differentially expressed genes (DEGs), gene dependencies, damaging mutations, and drug sensitivity data for NDNB-25-sensitive and resistant lines. Known Hsp90β interactors and curated drug–gene interactions from DGIdb were included to contextualize connections. (The visualization emphasizes global network structure and connectivity rather than individual node readability.) (**B**) Pathway enrichment of NDNB-25-resistant network. (**C**) Subnetwork derived from the broader NDNB-25 resistance network focusing on Rho GTPase signaling and actin cytoskeleton regulation. (**D**–**G**) ZIP synergy score surface plot and combination treatment bar plots for NDNB-25 and carboplatin combination in NDNB-25-resistant lines (**D**,**E**) and NDNB-25-sensitive lines (**F**,**G**). (**H**,**I**) ZIP synergy score plot and combination treatment bar plots for NCI-H23-R, an NDNB-25-acquired resistant cell line generated by passaging the parental line (NCI-H23) in escalating doses of NDNB-25. For all bar plots (panels (**E**,**G**,**I**)), statistical significance was assessed using one-way or two-way ANOVA followed by Dunnett’s multiple comparisons test and indicated as follows: ns = not significant (*p* > 0.05), *p* < 0.05 (*), *p* < 0.01 (**), *p* < 0.001 (***), *p* < 0.0001 (****).

## Data Availability

All data supporting the findings of this study are available within the article and its Appendix A. Publicly available datasets were obtained from the DepMap (https://depmap.org). Processed data and custom analyses were performed using the DepMap portal. These results were used to generate figures and perform pathway/network analyses, which are available in Appendix A. Additional processed data and analysis outputs are available from the corresponding author upon reasonable request. This study did not generate new unique reagents. Hsp90β inhibitors, including NDNB-25 and NDNB-21, are available upon reasonable request and with a material transfer agreement.

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
