# Peer review of "Integrative Multi-Omics Analyses Reveal Mechanisms of Resistance to Hsp90β-Selective Inhibition"

_cancers, 2025, doi:10.3390/cancers17213488_

Round 1

Reviewer 1 Report

Comments and Suggestions for Authors

This study presents a comprehensive mapping of the molecular landscape underlying resistance to Hsp90β selective inhibition, with a particular focus on the compound NDNB-25. The authors employ an integrative multi-omic strategy combining gene dependency, transcriptomics, metabolomics, and drug sensitivity data to dissect the complex resistance mechanisms involved.

Key findings include:

  • Resistance is mediated through cytoskeletal remodeling, metabolic adaptation, and stress response activation.
  • Critical regulators such as RHO GTPases, IQGAP1, ARHGDIA, VCL, and ANXA proteins are implicated in stabilizing cytoskeletal architecture under proteotoxic stress.
  • Resistant cells display metabolic reprogramming involving NAD⁺ metabolism, mitochondrial redox balance, and one carbon cycle intermediates.
  • A shift towards kynurenine pathway activation, characterized by increased kynurenine and decreased tryptophan, is suggested to drive AHR activation, potentially contributing to drug resistance.
  • While there is some overlap with genetic HSP90AB1 dependency, NDNB-25 resistance reveals distinct vulnerabilities, including altered Golgi trafficking and drug detoxification mechanisms.
  • Network analyses highlight convergence on RHO GTPase, PI3K/AKT, and receptor tyrosine kinase pathways (e.g., EGFR, FGFR1).
  • Interestingly, NDNB-25 resistant cells exhibit increased sensitivity to carboplatin, suggesting a potential therapeutic synergy between Hsp90β inhibitors and DNA-damaging agents.
  • Upregulation of Hsp90α in resistant cells suggests a compensatory response to Hsp90β inhibition.
  • The study effectively employs a flexible, multi-omic analytical framework that balances hypothesis-driven approaches with data guided exploration.
  • By prioritizing pathway and gene program level analysis over individual gene level observations, the authors aim to identify more generalizable and robust biomarkers of resistance.

The strategic utilization of existing large scale datasets (e.g., DepMap, CCLE) significantly enhances the depth and translational relevance of the study. Overall, this manuscript provides valuable insights into mechanisms of drug resistance and identifies potential therapeutic strategies that could benefit both the scientific community and cancer patients. I believe it represents a meaningful contribution to the field and supports further investigation into isoform selective Hsp90 inhibition and combination therapies.

Author Response

Reviewer 1 requires no response.

Reviewer 2 Report

Comments and Suggestions for Authors

The manuscript presents a thorough and timely study on mechanisms of resistance to Hsp90β-selective inhibition using multi-omics integration (transcriptomics, proteomics, metabolomics, gene dependency, and drug sensitivity) alongside experimental validation with NDNB-25 and NDNB-21. The authors highlight important pathways—Rho GTPase signaling, cytoskeletal remodeling, mitochondrial dysfunction, and kynurenine AHR signalling and propose carboplatin as a potential synergistic agent. Overall, the study is comprehensive and well-executed, but several points need clarification to strengthen mechanistic evidence, translational relevance, statistical rigor, and figure presentation.

Comments:

  1. The manuscript identifies key resistance mechanisms, but it would benefit from direct experimental validation. For example, can the authors show that kynurenine AHR signaling or compensatory Hsp90α upregulation is functionally driving resistance, perhaps using knockdown or pharmacological inhibition? Demonstrating causality would significantly strengthen the conclusions.
  2. Are the concentrations of NDNB-25 and NDNB-21 used in the experiments achievable in vivo? A small discussion of pharmacokinetics and clinical relevance would help contextualize the findings and the potential for therapeutic translation.
  3. The authors should clarify why a relatively lenient cutoff (adj. p < 0.2) was used in proteomic DEG analyses and whether multiple testing corrections were consistently applied across datasets. Strengthening the statistical interpretation of correlation and overlap analyses would also reduce reliance on descriptive observations.
  4. Key validation results—such as NDNB-21 responses, kynurenine/tryptophan correlations, and AHR-related evidence—should be moved from supplementary materials to the main figures. Network figures could also be simplified, reformatted, or presented in higher resolution to improve readability.
  5. Include more recent (2023–2024) references on Hsp90β-selective inhibitors in the introduction.
  6. Provide additional details on metabolomics processing (normalization, batch correction).
  7. Use “dependency” and “sensitivity” consistently throughout the text.

Author Response

Dear Editor,

We thank you and the reviewers for the thoughtful and constructive feedback on our manuscript. We have carefully addressed all comments through additional experimentation, clarifications, and targeted revisions, which have significantly strengthened both the mechanistic and translational aspects of the study. Below, we summarize the major revisions and highlight key improvements.

Response to Reviewer Comments

A detailed point-by-point response to reviewer’s comments is included here, describing how each comment has been addressed or, where direct experimentation was not feasible within the revision window, providing a clear explanation and plan for future validation.

References

All newly added citations directly relate to Hsp90β-selective inhibitor design, mechanism, or translational relevance. Reviewer-suggested or recent publications were carefully evaluated and included only where they substantively enhance the manuscript’s context. No irrelevant or redundant references were added.

Additional Notes

Where full resolution of certain mechanistic questions would more rigorous follow-up studies, we have clarified this limitation and outlined plans for future investigation.

All figures, tables, and supplementary materials have been updated accordingly, and all changes to the manuscript text have been highlighted.

Summary of Revisions

Reviewer 2

  1. The manuscript identifies key resistance mechanisms, but it would benefit from direct experimental validation. For example, can the authors show that kynurenine AHR signaling or compensatory Hsp90α upregulation is functionally driving resistance, perhaps using knockdown or pharmacological inhibition?

We appreciate the reviewer’s insightful suggestion and agree that additional functional validation strengthens our conclusions. To directly assess the kynurenine–AHR signaling axis, we performed LC-MS quantification of intracellular kynurenine in NDNB-25–acquired resistant and parental cell lines.  This analysis confirmed a significant elevation of kynurenine levels in resistant cells, supporting our model that AHR activation contributes to adaptive resistance. These new data have been incorporated into the revised manuscript (Results, Figure 4I).  Also, since the NDNB-25 acquired resistant line was originally introduced in results for Figure 6, we have now moved that data to Supplemental Figure 4E, and highlighted compensatory Hsp90α expression preceding the kynurenine validation in Figure 4G.  

The current study was intentionally designed as a broad exploratory characterization integrating transcriptomic, proteomic, metabolomic, and dependency datasets to identify convergent resistance mechanisms.  While comprehensive functional testing was beyond the present scope, the integrative analyses, drug-screening data, and kynurenine validation experiment together delineate several promising mechanistic leads.  These findings form the basis for ongoing context-specific studies, including lineage-focused mechanisms of resistance, kynurenine-AHR signaling and compensatory Hsp90α regulation in resistant models.

  1. Are the concentrations of NDNB-25 and NDNB-21 used in the experiments achievable in vivo? Provide pharmacokinetic/clinical context.

This is a good point and oversight from our original submission.  NDNB-25 and -21 compounds are part of a well-characterized series of Hsp90β-selective analogs, some of which have demonstrated in vivo efficacy and favorable safety profiles.  For example, the related compound NDNB1182 significantly enhanced immune checkpoint blockade therapy in murine tumor models without observable systemic toxicity (Frontiers Immunology. 2022; 13:1005045). These findings establish that Hsp90β-selective inhibition can achieve therapeutically relevant exposures in vivo with good tolerability, and we’ve added this to the discussion section.

  1. The authors should clarify why a relatively lenient cutoff (adj. p < 0.2) was used in proteomic DEG analyses and whether multiple-testing corrections were consistently applied across datasets.

Proteomic datasets typically exhibit greater technical variability and smaller feature counts than transcriptomic data, so we applied a permissive threshold (p < 0.2) to capture biologically meaningful trends while minimizing false negatives.  Additionally, for analyses related to differentially expressed genes, gene sets were restricted to genes that were significant in both proteomics data (relaxed cut-off) AND significant in the transcriptomics data with stricter thresholds (p < 0.05).  The Methods section (“Gene Expression and Proteomic Analysis,” now explicitly states this rationale and confirms consistent multiple-testing correction across all analyses.

  1. Key validation results—such as NDNB-21 responses, kynurenine/tryptophan correlations, and AHR-related evidence—should be moved from supplementary materials to the main figures.

We appreciate this suggestion and have reorganized key data to improve clarity and emphasis. The kynurenine correlation plot and AHR-related evidence have been moved from Supplementary Figure S4 to a new main-text figure (Figure 4G–I), where they are discussed in greater detail in the Results and Discussion sections. Because NDNB-21 produced similar but non-additive results relative to our current lead compound, NDNB-25, we retained those data as supporting evidence in the Supplementary section while clearly referencing them in the main text.

  1. Network figures could also be simplified, reformatted, or presented in higher resolution to improve readability.

Network figures (Figures 3 and 6) were reformatted for improved readability and exported at higher resolution.  Also note, network figures were generally used to visualize the methodology and broadly show connections between the separate feature analyses and not necessarily meant to focus on specific nodes.

  1. Include more recent (2023–2024) references on Hsp90β-selective inhibitors in the introduction.

We agree that inclusion of the most recent literature strengthens the Introduction.  We have incorporated several new citations describing the design, selectivity, and pharmacologic characterization of next-generation Hsp90β-selective scaffolds.

  1. Provide additional details on metabolomics processing (normalization, batch correction).

We revised the Metabolomics Analysis to describe Depmap/CCLE procedures citing relevant source material and added method details for the added revision data in regards to kynurenine intracellular LC–MS quantification.

  1. Use “dependency” and “sensitivity” consistently throughout the text.

The terms dependency and sensitivity are now used consistently to distinguish CRISPR/RNAi gene effect data (HSP90AB1-dependent and -resistant) from pharmacologic response measurements (NDNB-25-sensitive and -resistant).

We appreciate the opportunity to revise and resubmit our manuscript.  We believe these revisions have substantially improved the rigor, clarity, and translational impact of the work, and we hope it will now be suitable for publication.

Thank you for your consideration.

Sincerely,

Reviewer 3 Report

Comments and Suggestions for Authors

The manuscript is well presented, contains all the necessary information to understand what the authors wanted to do. It is well written, and the experimental design is appropriate. It is a good model to follow the search for new cancer markers. I suggest to publish in the present form but with the figures in high quality

Author Response

Reviewer 3 required no response.

Round 2

Reviewer 2 Report

Comments and Suggestions for Authors

The authors have done an excellent job revising the manuscript. They have addressed all my comments thoroughly and efficiently. The manuscript is now ready for publication, and I congratulate the authors on a well-executed and impactful study.